# TTC26/DYF13 is an intraflagellar transport protein required for transport of motility-related proteins into flagella

Hiroaki Ishikawa[1]*, Takahiro Ide[2†], Toshiki Yagi[3†], Xue Jiang[4†], Masafumi Hirono[2], Hiroyuki Sasaki[5,6‡], Haruaki Yanagisawa[2], Kimberly A Wemmer[1], Didier YR Stainier[1,7], Hongmin Qin[4], Ritsu Kamiya[2,8], Wallace F Marshall[1]*

[1]Department of Biochemistry and Biophysics, University of California, San Francisco, San Francisco, United States; [2]Department of Biological Science, Graduate School of Science, University of Tokyo, Tokyo, Japan; [3]Department of Cell Biology and Anatomy, Graduate School of Medicine, University of Tokyo, Tokyo, Japan; [4]Department of Biology, Texas A&M University, College Station, United States; [5]Division of Fine Morphology, Core Research Facilities, The Jikei University School of Medicine, Tokyo, Japan; [6]The Center for Advanced Medical Engineering and Informatics, Osaka University, Osaka, Japan; [7]Department of Developmental Genetics, Max Planck Institute for Heart and Lung Research, Bad Nauheim, Germany; [8]Department of Life Science, Faculty of Science, Gakushuin University, Tokyo, Japan

*For correspondence: hiroaki. ishikawa@ucsf.edu (HI); Wallace. Marshall@ucsf.edu (WFM)

[†]These authors contributed equally to this work

[‡]Present address: Department of Physical Therapy, Faculty of Community Health Care, Teikyo Heisei University, Chiba, Japan

Competing interests: The authors declare that no competing interests exist.

**Abstract** Cilia/flagella are assembled and maintained by the process of intraflagellar transport (IFT), a highly conserved mechanism involving more than 20 IFT proteins. However, the functions of individual IFT proteins are mostly unclear. To help address this issue, we focused on a putative IFT protein TTC26/DYF13. Using live imaging and biochemical approaches we show that TTC26/DYF13 is an IFT complex B protein in mammalian cells and *Chlamydomonas reinhardtii*. Knockdown of TTC26/DYF13 in zebrafish embryos or mutation of TTC26/DYF13 in *C. reinhardtii*, produced short cilia with abnormal motility. Surprisingly, IFT particle assembly and speed were normal in *dyf13* mutant flagella, unlike in other IFT complex B mutants. Proteomic and biochemical analyses indicated a particular set of proteins involved in motility was specifically depleted in the *dyf13* mutant. These results support the concept that different IFT proteins are responsible for different cargo subsets, providing a possible explanation for the complexity of the IFT machinery.

## Introduction

Cilia and flagella are hair-like microtubule-based organelles, which protrude from the cell surface. Cilia and flagella are basically similar structures and are present in organisms as diverse as single-celled eukaryotes and humans. Cilia have two major physiological functions. One function is producing a driving force for locomotion or making fluid flow (*Ostrowski et al., 2011*; *Vincensini et al., 2011*). The other function is sensing extracellular signals and environments, such as hedgehog signaling and fluid flow (*Goetz and Anderson, 2010*; *Drummond, 2012*). Because these ciliary functions are important for development and physiology, defects in cilia structure or function cause multiple human diseases (ciliopathies), such as primary ciliary dyskinesia, polycystic kidney disease, Bardet–Biedl syndrome, Meckel–Gruber syndrome, and Joubert syndrome (*Badano et al., 2006*; *Tobin and Beales, 2009*; *Hildebrandt et al., 2011*). Despite the importance of cilia, the mechanisms that assemble such complex structures are not fully understood.

The assembly and maintenance of cilia are known to be dependent on intraflagellar transport (IFT), an active transport process within cilia mediated by a bi-directional movement of multiprotein complexes,

**eLife digest** Sperm cells have tails called flagella that propel them towards an egg. Other cells have similar, but shorter, structures called cilia that sway back and forth on their surface. In addition to sweeping dust and debris out of our lungs and airways, cilia have a number of other crucial roles during development. This means that faulty cilia can lead to serious birth defects, as well as diseases of the kidneys and respiratory system.

Cilia and flagella are made from proteins that are assembled in a process called intraflagellar transport or IFT for short. Around 20 proteins are thought to be involved in this process, but the precise role of many of these proteins remains unclear. Now Ishikawa et al. have compared the versions of one of these proteins, called TTC26, that are found in zebrafish, mouse cells, and a single-celled alga called *Chlamydomonas reinhardtii* that uses a pair of flagella to move around.

This protein localizes to the cilia of mice cells and can be seen to move along these cilia in a manner typical of other IFT proteins. Ishikawa et al. then blocked production of TTC26 in zebrafish embryos, which caused these embryos to fail to develop the correct left–right asymmetry, and these fish also had problems with their eyes, ears, and kidneys. Furthermore and although cilia were present in the affected zebrafish, these cilia were shortened and moved abnormally. Ishikawa et al. also found that algae that had a mutation in the gene that codes for TTC26 had short cilia that moved in an abnormal way.

The findings of Ishikawa et al. suggest that TTC26 may help to transport a specific subset of proteins into the cilia. If other IFT proteins are also shown to carry distinct subsets of cargo, this might explain why as many as 20 different proteins are involved in the IFT process.

known as IFT particles, along the ciliary axoneme (*Kozminski et al., 1993*; *Rosenbaum and Witman, 2002*; *Pedersen et al., 2008*; *Scholey, 2008*; *Ishikawa and Marshall, 2011*). IFT complex movement is propelled by motor proteins, kinesin-2, and cytoplasmic dynein 2, which move toward the plus and minus ends of microtubules, respectively. Because proteins cannot be synthesized within the cilium, IFT is thought to be needed to carry ciliary components into cilia, by docking the cargo proteins onto the IFT complexes so that the cargo is carried along by the active movement of the complexes (*Piperno and Mead, 1997*; *Qin et al., 2004*; *Hao et al., 2011*). IFT complexes are composed of more than 20 proteins and motor proteins and can be separated biochemically and functionally into two subcomplexes, IFT complexes A and B (*Cole et al., 1998*). Why is the IFT system so complex? It is known that IFT complex B contributes to anterograde IFT with kinesin, and IFT complex A contributes to retrograde IFT with dynein. However, the functions of individual IFT proteins are mostly unclear. Because depletion of individual IFT complex proteins reduces the assembly of IFT particles and generally inhibits normal ciliogenesis or changes the morphology of the cilium (*Pazour et al., 2000*; *Brazelton et al., 2001*; *Deane et al., 2001*; *Tran et al., 2008*; *Mill et al., 2011*), it has been difficult to determine whether individual IFT proteins have specific functions other than IFT particle assembly. Specific functions of only a few IFT proteins have been identified. For example, IFT25 is important in transporting hedgehog signals, but is not required for cilia assembly (*Keady et al., 2012*). IFT46 is required for transport of outer dynein arms into flagella (*Hou et al., 2007*; *Ahmed et al., 2008*). IFT70/Fleer/DYF1 is involved in polyglutamylation of axonemal tubulin (*Pathak et al., 2007*; *Dave et al., 2009*). IFT172 contributes transition between anterograde and retrograde IFT at the tip of flagella (*Pedersen et al., 2005*). A recent study has demonstrated that the N-terminal parts of IFT74 and IFT81 form a tubulin-binding module (*Bhogaraju et al., 2013*). It is thus emerging that distinct IFT proteins may play distinct roles in transporting different sets of cargos in order to support different cilia functions; however, the cargo transport role of IFT proteins has not been restricted to testing individual candidate cargoes and has not employed systematic proteomic analyses.

In this study, we focused on TTC26/DYF13, which was identified in our proteomic analysis of mouse primary cilia (*Ishikawa et al., 2012*), as well as in other systematic studies of cilia (*Ostrowski et al., 2002*; *Avidor-Reiss et al., 2004*; *Li et al., 2004*; *Blacque et al., 2005*; *Efimenko et al., 2005*; *Stolc et al., 2005*; *Liu et al., 2007*; *Arnaiz et al., 2009*). TTC26 is a homologue of *Caenorhabditis elegans* DYF-13 (*Blacque et al., 2005*) and *Trypanosoma brucei* PIFTC3 (*Absalon et al., 2008*; *Franklin and Ullu, 2010*). The Dyf (dye-filling defective) phenotype reflects a defect in ciliary assembly in *C. elegans dyf-13*

mutants, and previous reports suggested that TTC26/DYF13 is a putative IFT protein (*Blacque et al., 2005*; *Absalon et al., 2008*; *Follit et al., 2009*; *Franklin and Ullu, 2010*). TTC26 knockdown has been reported to cause defects in the zebrafish retina and pronephros consistent with a ciliary defect (*Zhang et al., 2012*). In *C. elegans,* DYF-13 might be required to modulate the activation of kinesin-2 by docking this motor onto the IFT complex B (*Ou et al., 2005*, *2007*), but the exact function of the protein is still unknown.

To clarify the function of TTC26, we analyzed TTC26/DYF13 in mammalian cultured cells, zebrafish, and *Chlamydomonas reinhardtii*. Our results show that GFP-fused TTC26 moved bi-directionally along the length of cilia in mammalian cells just like other IFT proteins. TTC26/DYF13 was biochemically co-purified with other IFT complex B proteins in mammalian cells and in *C. reinhardtii*, demonstrating that TTC26/DYF13 is indeed an IFT complex B protein. However, unlike other complex B proteins, a deletion mutant of *dyf13* in *C. reinhardtii* still has flagella, indicating that TTC26/DYF13 is not required for ciliogenesis per se. However, the flagella are slightly shorter with pronounced motility defects. Similar phenotypes were observed in zebrafish cilia when *ttc26* was depleted. Although IFT particle movement seems normal in *dyf13* mutant flagella, specific inner dynein arm components were reduced in flagella of *dyf13* mutants, indicating a specific role for TTC26/DYF13 in transport of a subset of ciliary proteins. Comparative proteomic analysis of the flagellar proteins in the *dyf13* mutant supports a specific function for TTC26/DYF13 in transporting a subset of cargo proteins involved in the machinery of flagellar motility. Therefore, unlike many other IFT proteins, TTC26/DYF13 is not required for ciliogenesis nor for assembly or movement of the IFT particle per se, but rather plays a specialized role in transporting a specific set of motility-related ciliary proteins into cilia/flagella.

## Results

### TTC26 is a ciliary protein, which is conserved in ciliated organisms

In a previous report, we performed proteomic analysis of primary cilia from mouse cells and identified 195 proteins as candidate cilia proteins (*Ishikawa et al., 2012*). We compared identified proteins with other published systematic studies of cilia, such as proteomic, comparative genomic, and promoter analyses of cilia, in various organisms (*Ostrowski et al., 2002*; *Avidor-Reiss et al., 2004*; *Li et al., 2004*; *Blacque et al., 2005*; *Efimenko et al., 2005*; *Pazour et al., 2005*; *Stolc et al., 2005*; *Liu et al., 2007*; *Arnaiz et al., 2009*; *Ishikawa et al., 2012*), and found that some proteins in our candidate list had also been identified in other types of systematic studies of cilia. These proteins are presumably of particular importance to the function or assembly of cilia. We focused on one of the identified proteins, TTC26, which is a homologue of *C. elegans* DYF-13 (*Blacque et al., 2005*) and *T. brucei* PIFTC3 (*Absalon et al., 2008*; *Franklin and Ullu, 2010*). TTC26 has tetratricopeptide repeat motifs and is highly conserved not only in vertebrates but also in a variety of ciliated organisms (*Figure 1A*). Orthologs were not found in nonciliated organisms. Mouse TTC26 shows 58% amino acid identity and 73% similarity with the *C. reinhardtii* homologue. Endogenous mouse TTC26, detected by antibody staining, exclusively localized to primary cilia and basal bodies in IMCD3 cells, which derived from mouse kidney collecting ducts (*Figure 1B*). Endogenous TTC26 showed punctate localizations along the length of the primary cilium, resembling the pattern usually seen with IFT proteins.

### Knockdown of TTC26 in zebrafish shows phenotypes typical of defective cilia

Because cilia are important in development (*Drummond, 2012*; *Oh and Katsanis, 2012*), we investigated TTC26 function during zebrafish embryogenesis. The mRNA of *ttc26* is present in ciliated tissues in zebrafish, such as Kupffer's vesicle (KV) and pronephric ducts (http://zfin.org). To understand the function of TTC26 in vertebrate development, we disrupted the function of the zebrafish *ttc26* gene by injecting an antisense morpholino. We used translational blocking and splice blocking morpholinos for knocking down *ttc26*. Both morpholinos effectively suppressed TTC26 expression (*Figure 2—figure supplement 1A* ) and caused phenotypes that are typical of defective cilia in zebrafish (*Malicki et al., 2011*), including left–right asymmetry defects with abnormal heart looping (*Figure 2A,B*), as well as hydrocephalus, pronephric cysts, abnormal ear otolith formation, and curly body axis (*Figure 2—figure supplement 1B,C*). In *Figure 2A*, the heart-looping is normally oriented to rightward. The appearance of the ventricle on the left of the atrium indicates a left–right reversal of the normal heart-looping pattern. The pronephric cyst and curled body axis phenotypes were

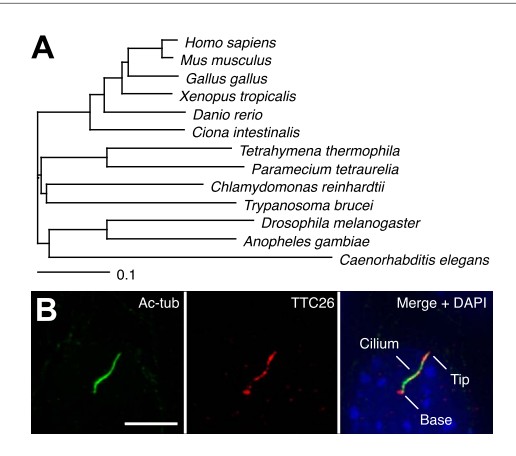

**Figure 1**. TTC26 is a conserved ciliary protein. (**A**) Phylogenetic tree of TTC26 in various ciliated organisms. All protein sequences were obtained from RefSeq (NCBI). The scale bar represents 0.1 substitutions per nucleotide site. (**B**) Immunofluorescence images of primary cilia in IMCD3 cells. The cells were stained with antibodies to acetylated tubulin (green), TTC26 (red) and DAPI (blue). Bar: 5 μm.

previously reported by *Zhang et al. (2012)*. Many of these phenotypes also share similarities with human ciliopathies (*Badano et al., 2006*; *Tobin and Beales, 2009*; *Hildebrandt et al., 2011*). In light of the prior report that *ttc26* morphant zebrafish had photoreceptor defects in the retina (*Zhang et al., 2012*), we checked eye histology by transmission electron microscopy to examine the specialized connecting cilium of the photoreceptor. *ttc26* morpholino-injected animals (morphants) had lost most of the outer segments of their photoreceptors as previously reported by *Zhang et al. (2012)*. However, we found that photoreceptor cells in *ttc26* morphant animals still had a connecting cilium (*Figure 2—figure supplement 1D,E*), indicating that TTC26 may not be essential for connecting cilia maintenance but might instead affect their function. However, we note it is formally possible that maternally supplied TTC26 protein might support connecting cilium assembly. Altogether, these data show that TTC26 is important for cilia function, not only in the retina but also throughout zebrafish development.

## TTC26 morphants have short cilia with motility defects

The fact that photoreceptor cilia are still present in *ttc26* morphant zebrafish suggests that the effect on cilia may be more subtle than a frank failure of ciliogenesis. To address the role of TTC26 in ciliogenesis, we visualized cilia in whole mount-fixed embryos by immunohistochemistry with acetylated tubulin antibody. We checked cilia length in the KV, which is a ciliated organ in the zebrafish embryo that is essential to mediate left–right asymmetry (*Essner et al., 2005*). Cilia in the KV are well separated from each other and hence particularly easy to measure by light microscopy, compared to either the tiny cilia of the spinal cord or the close-packed cilia of the pronephros. Cilia were present in the KV of the *ttc26* morphants, but the mean length of cilia in the KV at the 10-somite stage was significantly reduced (control 6.4 ± 2.4 μm, *ttc26* morphants 4.4 ± 1.6 μm; *Figure 2C,D*). Moreover, the mean number of cilia in the KV was also reduced in *ttc26* morphants (control 51.1 ± 17.3, *ttc26* morphants 28.1 ± 11.1; *Figure 2E*). *ttc26* morphants also exhibited shorter cilia in pronephric ducts than control fish at the 22-somite stage (control 3.8 ± 1.2 μm, *ttc26* 2.8 ± 0.9 μm; *Figure 2F,G*). These results show that TTC26 is important to assemble full-length cilia in zebrafish.

The cilia in the KV are motile and generate the directional fluid flow that is crucial to break early bilateral symmetry (*Essner et al., 2005*; *Neugebauer et al., 2009*). To assess whether cilia-driven directional fluid flow in the KV was altered by the cilia defects in *ttc26* morphants, we injected fluorescent beads into the lumen of the KV and tracked their movement. In control morphants, fluorescent beads flowed in a persistent counter-clockwise direction (*Figure 2H*; *Video 1*). In contrast, this persistent directional flow was abolished in *ttc26* morphants (*Figure 2H*; *Video 2*). We also directly visualized individual cilia motility in vivo within the KV by high-speed microscopy using differential interference contrast (DIC) optics. In control morphants, motile cilia displayed a circular beat with a mean frequency of 28.9 ± 6.3 Hz (*Figure 2I,J*; *Video 3*). Cilia of *ttc26* morphants also showed a circular beat in most cases, but in contrast to control morphants, beat with a significantly reduced mean frequency of 19.2 ± 4.4 Hz, and sometimes showed unusual beating motion (*Figure 2I,J*; *Video 4*). These findings demonstrate that TTC26 is important for cilia motility.

## *C. reinhardtii dyf13* mutants show flagella motility defects

Zebrafish studies showed that TTC26/DYF13 plays important roles in development and physiology consistent with a role in ciliary function and full-length assembly. To further clarify the function of the protein at a cellular or molecular level, we isolated a *C. reinhardtii* mutant in the *DYF13* gene

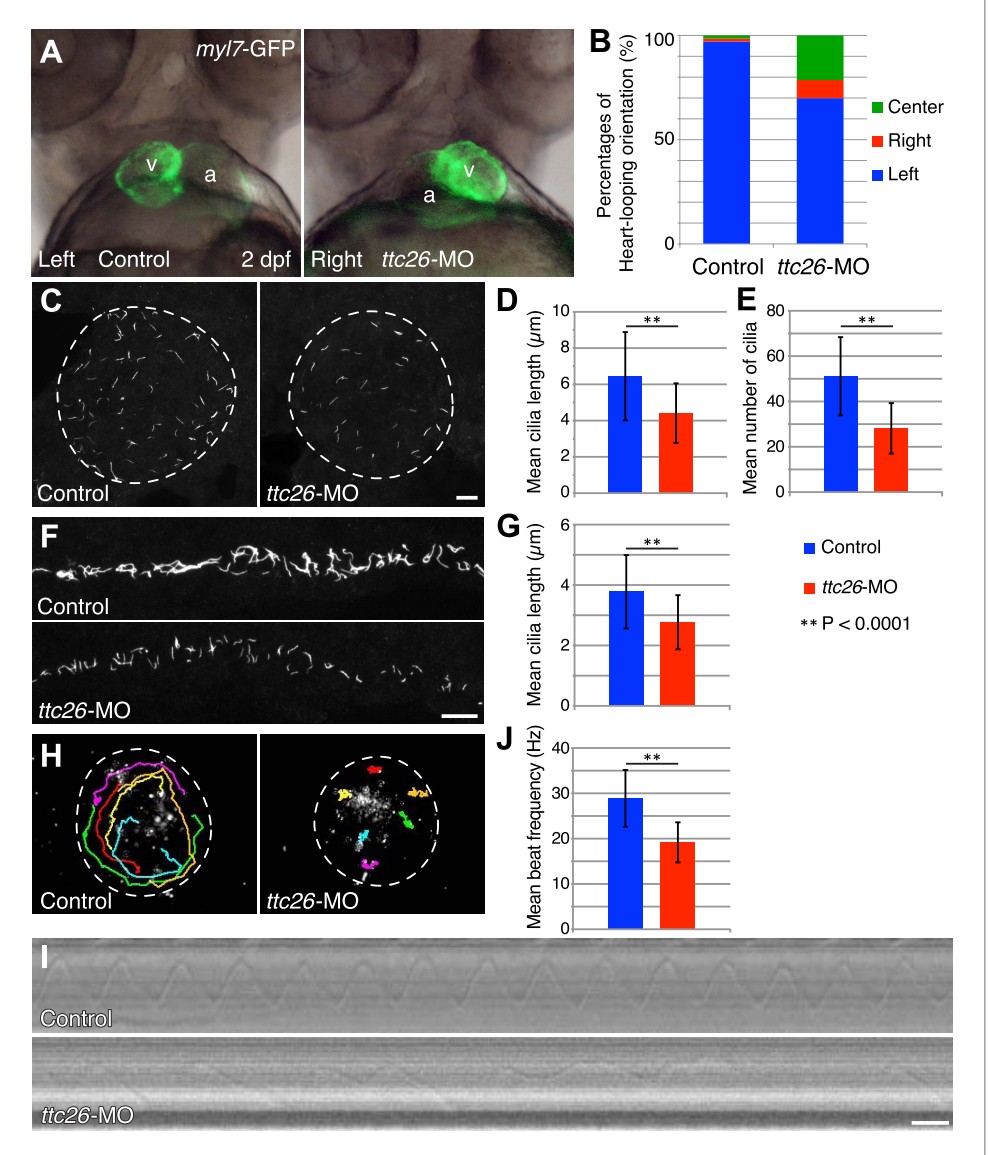

**Figure 2**. Knockdown of *ttc26* in zebrafish induces shorter cilia and motility defect. (**A**) Representative images of heart-looping orientations of control (Left: normal) and *ttc26* knockdown (Right: reversed) fish at 2 dpf. *myl7*-GFP transgenic embryos were injected with control or *ttc26* morpholinos. Atrium and ventricle are indicated by the letters a and v. (**B**) Percentages of heart-looping orientation of control (n = 338) and *ttc26* (n = 309) knockdown fish. Around 97% of control morphants showed normal heart-looping orientations (left: blue). In contrast, more than 30% of *ttc26* morphants showed abnormal heart-looping orientations (right: red and center: green). (**C**) Immunofluorescence images of Kupffer's vesicle in control and *ttc26* knockdown embryos, which were stained with antibodies to acetylated tubulin at 10-somite stage. Dashed line shows boundary of KV. Bar: 10 μm. (**D**) Length of KV cilia in control (n = 920) and *ttc26* knockdown fish (n = 506, p<1 × 10$^{-69}$). Blue shows control and red shows *ttc26* knockdown embryo. Error bars represent standard deviations in all figures. (**E**) Number of KV cilia in control (blue, n = 18) and *ttc26* knockdown fish (red, n = 18, p<1 × 10$^{-4}$). (**F**) Immunofluorescence images of pronephric ducts that were stained with antibodies to acetylated tubulin at the 22-somite stage. Bar: 10 μm. (**G**) Length of pronephric duct cilia in control (n = 103) and *ttc26* knockdown fish (n = 75, p<1 × 10$^{-8}$). (**H**) The traces of fluorescent beads in KV in control and *ttc26* knockdown embryos at the 8-somite stage showing impaired fluid flow in the morphant. Six representative tracks are shown in different colors in each image. These tracks were traced from the first 50 frames of *Videos 1 and 2*. (**I**) Kymographs of individual KV cilia in control and *ttc26* knockdown embryos. These kymographs were assembled from *Videos 3 and 4*. Bar: 20 ms.
*Figure 2. Continued on next page*

*Figure 2. Continued*

(**J**) Beat frequency of KV cilia in control (blue, n = 38) and *ttc26* knockdown fish (red, n = 21, p<1 × 10⁻⁸). Error bars show standard deviations.

The following figure supplements are available for figure 2:

**Figure supplement 1**. *ttc26* knockdown zebrafish shows phenotypes typical of defective cilia.

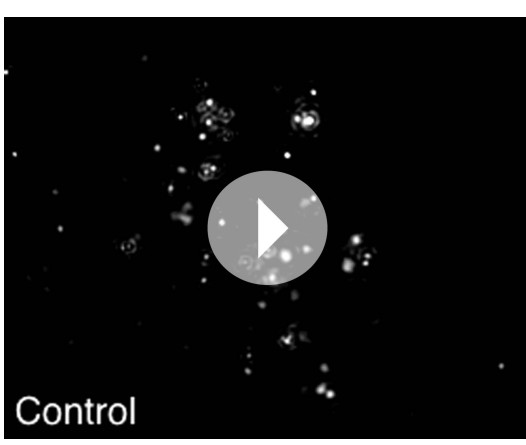

**Video 1**. Fluorescent videomicroscopy of fluid flow in the KV of control embryos. A dorsal view of the KV from a live control embryo at 8–10-somite stage. Fluorescent beads were microinjected into the KV. Images were collected at 5 fps for a total duration of 1 min. Playback is set at 20 fps (4 × speed). Beads show counter-clockwise movements in the KV.

**Video 2**. Fluorescent videomicroscopy of fluid flow in the KV of *ttc26* knockdown embryos. A dorsal view of the KV from a live *ttc26* knockdown embryo at 8–10-somite stage. Fluorescent beads were microinjected into the KV. Images were collected at 5 fps for a total duration of 1 min. Playback is set at 20 fps (4 × speed). Beads do not show directional movements.

(as the TTC26 homologue has been annotated in the *C. reinhardtii* genome). Such a mutant provides a key tool to clarify the molecular function of TTC26/DYF13 because it lets us exploit the advantages of *C. reinhardtii* as a model organism for biochemical and cellular studies of cilia and flagella. To obtain such a mutant, we checked the previously uncharacterized *C. reinhardtii* mutant stocks from our UV-mutagenesis screen conducted for flagellar motility defective mutants and found one mutant with short flagella and a motility defect which carries two mutations in the *DYF13* gene, a one-base substitution (c.57C>T) and a one-base deletion (c.59delC; *Figure 3A*). The product of the *dyf13* mutant gene is predicted to encode the correct first 19 amino acids and incorrect subsequent 20–56 amino acids (*p.Ala20Glufs*; *Figure 3B*) and is thus unlikely to retain any normal biochemical function. In contrast to wild-type *C. reinhardtii* cells in which DYF13 protein is clearly visible in puncta distributed along the length of the flagellum (*Figure 3C*), *dyf13* mutant cells lacked detectable DYF13 in their flagella (*Figure 3D*) and had significantly shorter flagella (8.1 ± 1.2 µm) than wild-type flagella (11.4 ± 0.9 µm; *Figure 3E,F*). In vegetative cells during logarithmic growth, mutant cells swam more slowly (25.1 ± 6.0 µm/s; *Figure 3G*, red) than wild-type cells (123.1 ± 13.7 µm/s; *Figure 3G*, blue). The flagellar beat frequency (25.9 ± 6.1 Hz; *Figure 3H*, red; *Video 5*) was also reduced compared to the wild-type frequency (47.1 ± 4.7 Hz; *Figure 3H*, blue; *Video 6*). Expression of HA-tagged DYF13 rescued these phenotypes (data not shown), confirming the mutation in DYF13 was the cause of the phenotypes. Thus, the shorter flagellar length and reduced beating frequency of the *dyf13* mutant were consistent with that of zebrafish *ttc26* morphants. Thus, in both zebrafish and *C. reinhardtii*, TTC26/DYF13 is required for assembly of full-length cilia with normal functions.

One might imagine that the length defect could be the cause of the motility defect if shorter flagella were impaired in their ability to generate a proper flagellar beat, however this does not appear to be the case. We performed identical motility assays on shorter flagella in wild-type cells that are regenerating their flagella after pH shock. Their short

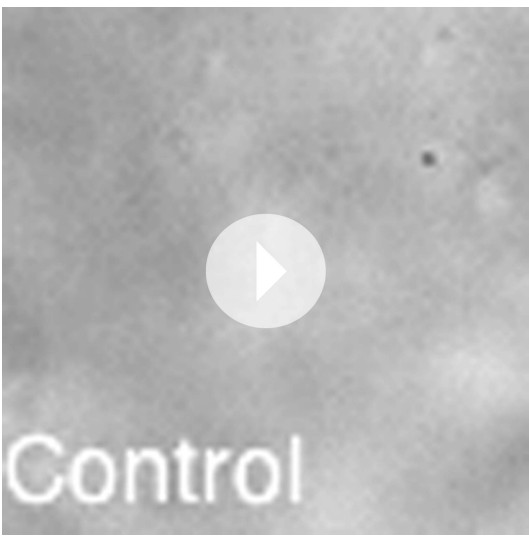

**Video 3**. High-speed DIC videomicroscopy of KV cilia from control embryos. A dorsal view of the KV from a live control embryo at 8–10-somite stage. DIC imaging was performed on an inverted microscope with a high-speed camera. Images were collected at 1000 fps for a total duration of 1 s. Playback is set at 20 fps (1/50 speed). The cilia show a circular motion with a mean frequency of 29.4 ± 7.4 Hz.

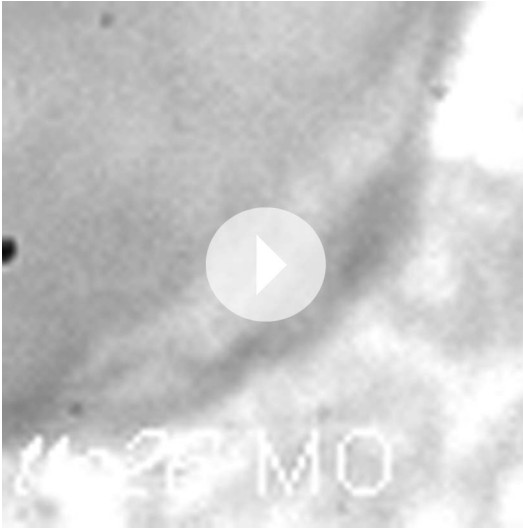

**Video 4**. High-speed DIC videomicroscopy of KV cilia from *ttc26* knockdown embryos. A dorsal view of the KV from a live *ttc26* knockdown embryo at 8–10-somite stage. DIC imaging was performed on an inverted microscope with a high-speed camera. Images were collected at 1000 fps for a total duration of 1 s. Playback is set at 20 fps (1/50 speed). The cilia show a circular motion but with a slower mean frequency of 17.7 ± 3.3 Hz and sometimes showed unusual beating motion.

flagella showed faster beat frequency (57.6 ± 6.7 Hz; *Figure 3H*, light blue) and slower swim speed (54.6 ± 12.5 µm/s; *Figure 3G*, light blue) than normal length flagella (*Figure 3G,H*, blue). This result is the opposite of what is seen in the *dyf13* mutant in which beat frequency is slower, not faster, than wild-type, and suggests that the reduced length of *dyf13* flagella is not the direct cause of the reduced beat frequency in the mutant. This result is also consistent with prior studies of short-flagella *C. reinhardtii* mutants which are reported to have normal flagella beating (*Jarvik et al., 1984*; *Kuchka and Jarvik, 1987*). These results and fact suggest that DYF13 has a role in building a motile flagellum above and beyond its role in flagellar length.

## TTC26/DYF13 is an IFT complex B protein

Given our results that TTC26/DYF13 has conserved roles in building full-length cilia and flagella in both zebrafish and *C. reinhardtii*, and given the fact that DYF-13 protein undergoes IFT motion in *C. elegans* (*Blacque et al., 2005*), we analyzed the dynamics of TTC26/DYF13 in mammalian cilia to determine if TTC26 undergoes IFT-like movements. We expressed GFP-tagged mouse TTC26 protein in mouse cultured cells and observed the behavior of TTC26-GFP in living cells using total internal reflection fluorescence (TIRF) microscopy. TTC26-GFP showed bi-directional movement in the mammalian cilium (*Figure 4A*, top; *Video 7*). The mean anterograde speed of mouse TTC26-GFP was 1.22 ± 0.17 µm/s, and the mean retrograde speed was 0.92 ± 0.24 µm/s (*Figure 4B*, blue). These speeds are almost the same as the speeds of known IFT proteins, such as IFT88 (*Figure 4A*, bottom and 4B, red; *Video 8*), suggesting that TTC26 is likely to move with other IFT proteins.

We next analyzed TTC26 protein interactions by expressing a tandem affinity purification (TAP)-tagged construct of mouse TTC26 in mouse IMCD3 cells. Using affinity purification via the TAP-tag, we pulled out a set of proteins that physically interact with TTC26 (*Figure 4C*) and then determined their identity by mass spectrometry ('Materials and methods'). This TAP analysis identified all known IFT complex B proteins, but no IFT complex A proteins nor motor proteins (*Figure 4C*; *Supplementary file 1A*).

To confirm that TTC26/DYF13 is an IFT complex protein by a different biochemical method, we performed sucrose density gradient analysis of membrane-matrix proteins isolated from *C. reinhardtii* flagella (*Cole et al., 1998*; *Wang*

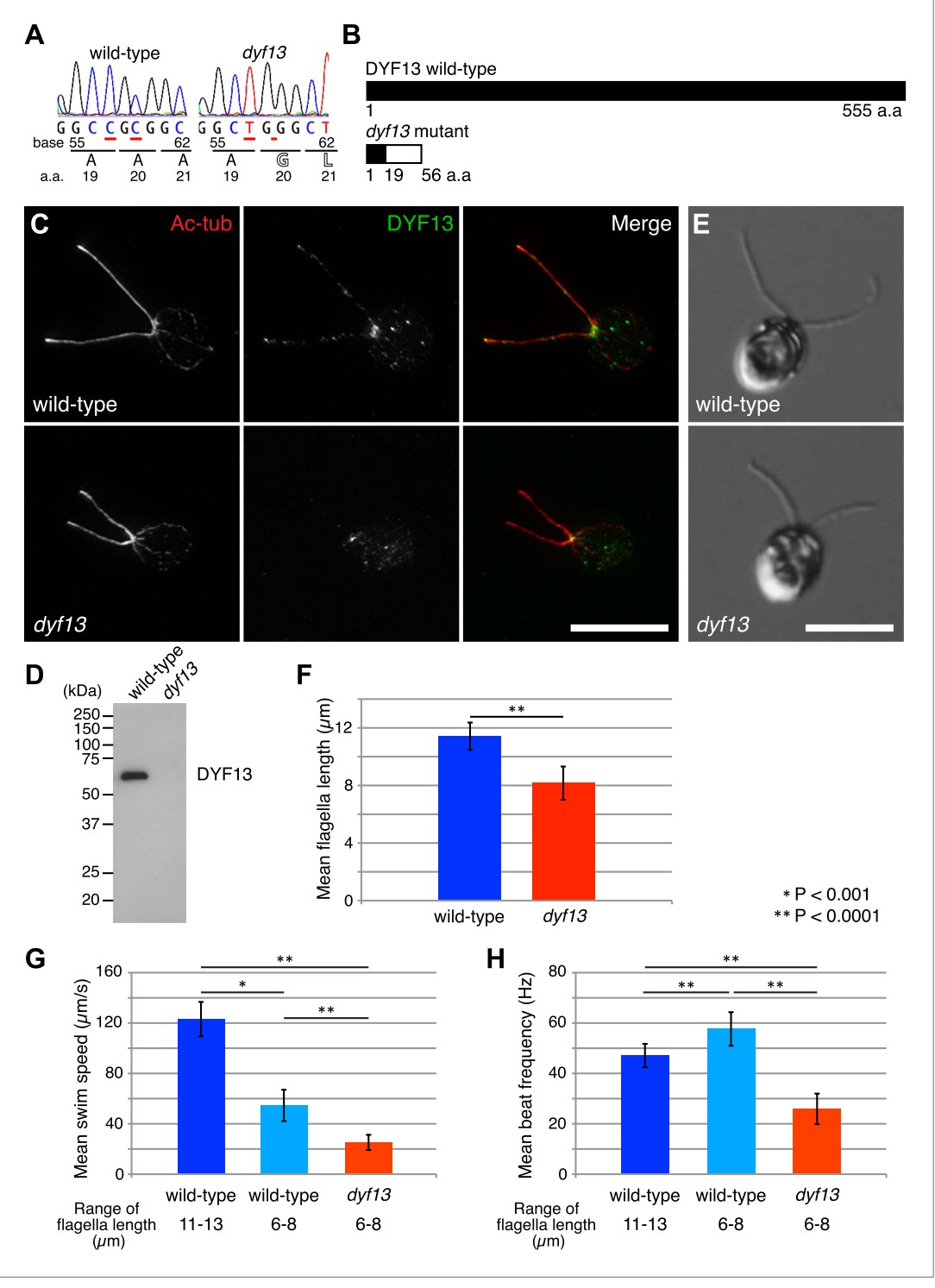

**Figure 3**. *C. reinhardtii dyf13* mutant has short flagella and motility defects. (**A**) Sequences of the *DYF13* gene (RefSeq accession: XM_001698717) in wild-type (CC125) and *dyf13* mutant cells. Red underlines show places of mutations. A one-base substitution (c.57C >T) does not change amino acid. A one-base deletion (c.59delC) in the *dyf13* gene induces frame-shift after 20th amino acid (a.a.) reside (*p.Ala20Glufs*). (**B**) Schematic representation of *C. reinhardtii* DYF13 (XP_001698769) and its mutant. The black bar indicates correct amino acid sequence and the white bar indicates frame-shift sequence. (**C**) Immunofluorescence images of wild-type and *dyf13* mutant cells. The cells were stained with antibodies to acetylated tubulin (red) and DYF13 (green). Bar: 10 μm. (**D**) Isolated flagella
*Figure 3. Continued on next page*

*Figure 3. Continued*

fractions from wild-type cc125 and *dyf13* mutant cells were analyzed in Western blots probed with the DYF13 antibody. The DYF13 antibody specifically recognized a ~56 kDa band. (**E**) DIC images of wild-type and *dyf13* mutant cells. Bar: 10 µm. (**F–H**) Flagella length (**F**), swim speed (**G**), and beat frequency of flagella (**H**) of wild-type (blue, n = 12) and *dyf13* mutant cells (red, n = 12, p<1 × 10⁻⁶, 10⁻⁸, 10⁻¹², respectively). Swim speeds and beat frequencies of flagella were measured also in wild-type cells with short flagella, which were adjusted by pH shock and regeneration (light blue, n = 10). Error bars represent standard deviations.

*et al., 2009*). The gradient pattern clearly showed that TTC26/DYF13 comigrated with other IFT complex B proteins, such as IFT46 and IFT74, but not IFT complex A protein (**Figure 5A**). These results demonstrate that TTC26/DYF13 is an IFT complex B protein, in both vertebrates and *C. reinhardtii*.

## TTC26/DYF13 does not influence IFT particle assembly or movement

Having found that TTC26/DYF13 is part of IFT complex B, we next asked whether it is required for the complex to form. We performed sucrose density gradient analysis on flagellar extracts from *C. reinhardtii dyf13* mutant flagella. Although the peak of IFT complex B in the *dyf13* mutant shifted slightly to a lighter density in the gradient, the IFT complex B proteins still comigrated with each other (**Figure 5B**).

Since our imaging and biochemical results indicated that IFT complex B still forms in the *dyf13* mutant, we next asked whether DYF13 is required for the movement of IFT particles. We constructed a *C. reinhardtii dyf13* mutant line expressing a GFP-tagged KAP subunit of the IFT kinesin-2, in a *fla3* mutant background that lacks the endogenous KAP protein (**Mueller et al., 2005**). Using this strain, we visualized KAP-GFP signals by TIRF microscopy in living cells (**Figure 6A**, top and 6B, blue; **Video 9**). First of all, we found that IFT particles could be clearly seen to move, indicating that DYF13 is not required for assembling IFT particles. The anterograde IFT speed of KAP-GFP in *dyf13* mutant cells was found to be slightly slower than control cells (**Figure 6A**, bottom and 6B, red; **Video 10**). However, interpreting this speed reduction is complicated by the fact that IFT speed is known to change as a function of flagellar length (**Engel et al., 2009**; **Besschetnova et al., 2010**), such that shorter flagella exhibit slower IFT speed in *C. reinhardtii* (**Engel et al., 2009**). Therefore, to determine if the *dyf13* mutation has a direct effect on IFT speed, we exploited the natural variation in flagellar lengths in a population of cells, and measured the KAP-GFP speed specifically in a subset of control cells that had the same range of flagellar lengths seen in the *dyf13* mutant. We found that in such cells, which were genetically wild-type but happened to have shorter flagella comparable to the *dyf13* mutant flagella (6–8 µm), the KAP-GFP speed was almost the same as in the *dyf13* mutant (**Figure 6B**, light blue). This result is consistent with the idea that the reduction of IFT speed in the *dyf13* mutant is an indirect consequence of the reduced flagellar length. The frequency of IFT trains as visualized by KAP-GFP did not change between control and *dyf13* mutant cells in the same range of flagella length (**Figure 6C**). We also checked the speed and frequency of another IFT complex B component, IFT27-GFP. In *dyf13* mutant flagella, IFT27-GFP showed almost the same speed and frequency as control for both anterograde and retrograde trains in the same range of flagella length (**Figure 6D–F**).

We also checked the localization and amount of IFT proteins in the *dyf13* mutant cells by immunofluorescence microscopy and Western blotting. The defect of the *dyf13* gene did not change the localization or amount of other IFT proteins (**Figure 7**).

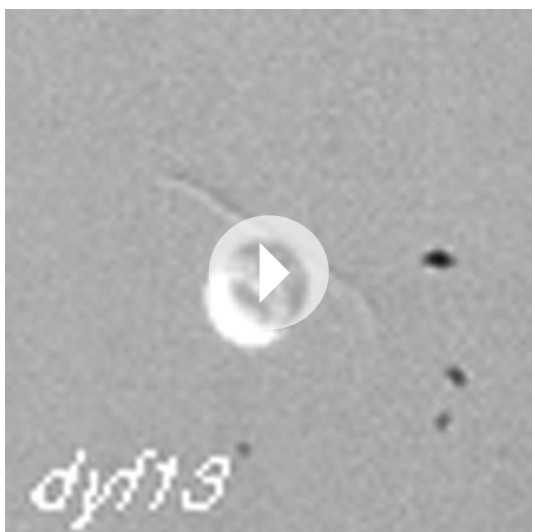

**Video 5**. High-speed DIC videomicroscopy of *C. reinhardtii dyf13* mutant cell. DIC imaging was performed on an inverted microscope with a high-speed camera. Images were collected at 1000 fps for a total duration of 0.16 s. Playback is set at 20 fps (1/50 speed). The *dyf13* mutant cell swims significantly slower than wild-type and flagellar movements are uncoordinated.

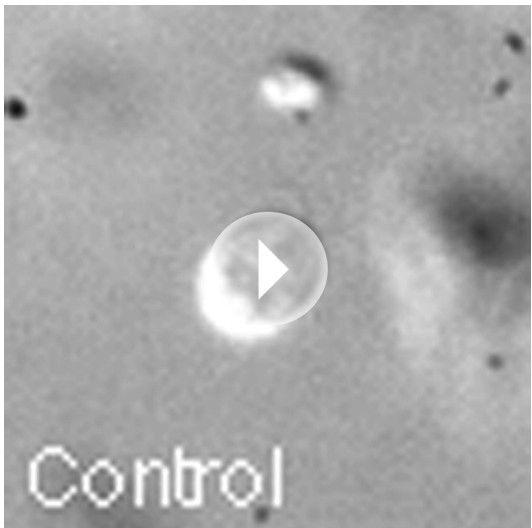

**Video 6**. High-speed DIC videomicroscopy of *C. reinhardtii* wild-type cell. DIC imaging was performed on an inverted microscope with a high-speed camera. Images were collected at 1000 fps for a total duration of 0.16 s. Playback is set at 20 fps (1/50 speed). The wild-type cell swims smoothly and both flagella beat together in a synchronized breast-stroke motion.

Taken together, these results indicate that DYF13 does not significantly influence the IFT particle assembly or movement.

## TTC26/DYF13 is important to transport inner dynein components into flagella

Defects of DYF13 did not change IFT speed or frequency even though DYF13 is an IFT protein. However, *C. reinhardtii dyf13* mutant cells clearly showed the flagella motility defects. These results raised the possibility that the DYF13 protein might be important for transporting specific cargo into the flagellum that is required for normal flagellar motility. Because the motility of flagella is mainly produced by the two types of dynein arms, inner and outer (*King and Kamiya, 2009*; *Wirschell et al., 2011*), we examined components of axonemal dyneins in *dyf13* mutant flagella. The outer dynein components from isolated axonemes of wild-type and *dyf13* mutant were purified and quantified by SDS-PAGE. Components of outer dynein arms were present at normal levels in *dyf13* mutant flagella (*Figure 8—figure supplement 1A,B*). To analyze the inner dynein arm components, the *dyf13* mutant was crossed with the *oda1* mutant, which are missing outer dynein arms (*Kamiya, 1988*). Inner-arm dyneins were purified from the high salt extract of *oda1dyf13* double mutant flagella by ion-exchange column chromatography on a Mono Q column. The elution pattern of the *oda1dyf13* axonemal extract indicated a reduction in the amounts of specific inner-arm dyneins (*Figure 8A*). The amounts of inner-arm dynein a, f, and g in *oda1dyf13* double mutant flagella were reduced 50% compared to the *oda1* single mutant (*Figure 8B*). This result indicates that DYF13 is important to transport inner dynein components into flagella and suggests that the loss of these dyneins causes flagella motility defects in the *dyf13* mutant.

## Proteomic analysis of *C. reinhardtii dyf13* mutant flagella

The loss of a specific set of inner dynein arm proteins suggested that TTC26/DYF13 has a role in transporting specific cargo proteins. To identify other potential cargos of DYF13, we performed a comparative proteomic approach using 2D-DIGE (two-dimensional difference in gel electrophoresis). Isolated flagella from wild-type and *dyf13* mutant cells were labeled by a different fluorescent dye, respectively, and were simultaneously separated on a single 2D gel, using isoelectric focusing and SDS-PAGE (*Figure 8C*). Out of 2324 distinct spots that could be detected on the gels, only 108 showed a significant change in abundance in mutant vs wild-type flagella as judged by an alteration of 1.5× or higher. Of these, we are primarily interested in those showing a reduction in abundance in the mutant compared to wild-type. Two separate experiments, in which flagella were prepared separately from independent biological replicate samples and then analyzed by DIGE, gave a similar pattern of spots for which abundance was reduced by 1.5× or more (64 out of 2324 in the first experiment compared to 106 out of 3358 in the second), with 53 of the spots from the first experiment also showing reduced abundance in the second. Evidently, the vast majority of flagellar proteins do not require TTC26/DYF13 for their import into flagella, suggesting that TTC26/DYF13 only transports a small subset of proteins. To better understand the nature of this subset, we picked a random sample of spots from the set of spots which showed at least a 1.5-fold difference in fluorescent intensities between wild-type and *dyf13* mutant cells, and identified the corresponding proteins by mass spectrometry. This was done for 16 spots that showed abundance changes in both DIGE experiments, and corresponding spots from both DIGE gels were extracted and analyzed by mass spectrometry. All 16 spots allowed

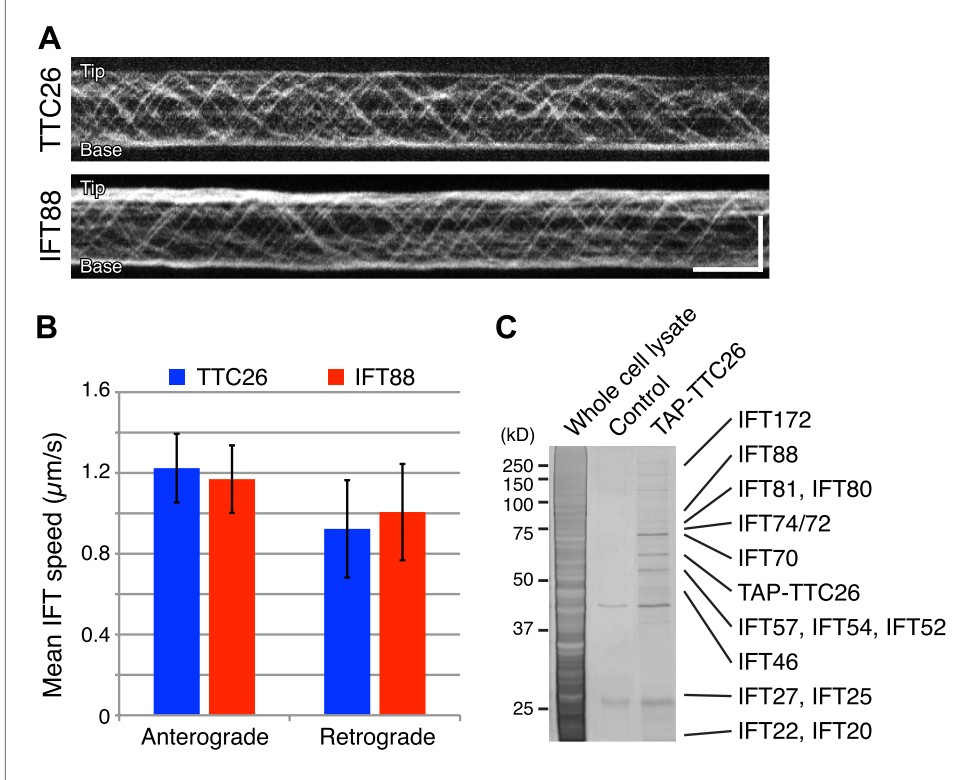

**Figure 4**. TTC26/DYF13 is an IFT complex B protein. (**A**) Kymographs of GFP-tagged mouse TTC26 and IFT88 moving in cilia of mouse IMCD3 cells. These kymographs were assembled from **Videos 7 and 8**. Bars: 10 s (horizontal), 5 μm (vertical). (**B**) The mean IFT speeds of TTC26- and IFT88-GFP. TTC26-GFP (blue, n = 37 cilia, more than 200 particles) and IFT88-GFP (red, n = 43 cilia, more than 200 particles) showed similar speeds in both anterograde and retrograde movements. Error bars represent standard deviations. (**C**) TTC26/DYF13 physically associates with IFT complex B proteins in mouse cilia. Proteins obtained from tandem affinity purification of TTC26 were separated on an acrylamide gel and visualized by silver staining as shown. The identity of each protein bands as determined by mass spectrometry is indicated next to the band, with details given in **Supplementary file 1A**.

protein identification from the mass spectra, as listed in **Supplementary file 1B**. The mass spectrometry analysis in the two separate experiments gave identical results for both experiments for each spot. Abundance changes were correlated between the two experiments (r = 0.68; p=0.0013). We confirmed that one of these proteins, centrin, showed a similar reduction in abundance using Western blotting (**Figure 8—figure supplement 1C**).

Most of the proteins reduced in *dyf13* mutant flagella (shown as a negative value in **Supplementary file 1B**) have been described in the literature as being involved in flagella motility, including FAP59, Enolase, RIB72, PF2, and centrin (see 'Discussion' for details). This result indicates that DYF13 is important to transport other components of motility regulatory machineries, such as the dynein regulatory complex and the central pair complex, as well as inner dynein arms. Interestingly, LIS1, a dynein regulatory factor which is associated with flagellar dynein arms (**Pedersen et al., 2007**), accumulated in *dyf13* mutant flagella. Because LIS1 accumulates in flagella when flagellar motility is arrested (**Rompolas et al., 2012**), the accumulation may be a secondary effect of flagella motility defects in *dyf13* mutant.

## Discussion

In this study, we analyzed TTC26/DYF13 in mammalian cells, zebrafish, and *C. reinhardtii* and found that TTC26/DYF13 is an IFT complex B protein but, unlike other IFT B proteins, it is not required for IFT complex B assembly or motion or for ciliogenesis in general, but rather is specialized for transport of a specific set of ciliary cargo proteins related to motility. We suggest calling this protein IFT56 based

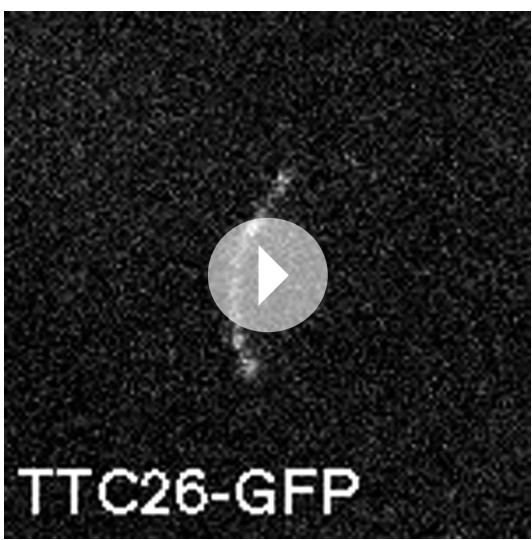

**Video 7**. Fluorescent videomicroscopy of TTC26-GFP in a cilium of IMCD3 cells. Images were collected at 10 fps for a total duration of 1 min. Playback is set at 20 fps (2 × speed). TTC26-GFP moves bi-directionally along the cilium. Orientation of cilia in these movies was determined using the position of the nucleus as a reference for the proximal end of the cilium.

**Video 8**. Fluorescent videomicroscopy of IFT88-GFP in a cilium of IMCD3 cells. Images were collected at 10 fps for a total duration of 1 min. Playback is set at 20 fps (2 × speed). IFT88-GFP moves bi-directionally along the cilium.

on its molecular weight (*Figure 3D*), following standard nomenclature for IFT proteins (*Cole et al., 1998*; *Rosenbaum and Witman, 2002*).

## IFT56 is not necessary for assembly or movement of IFT particles

We demonstrated that IFT56/TTC26/DYF13 is an IFT complex B by biochemical and cell biological analyses. IFT56 comigrates with IFT complex B on sucrose gradients (*Figure 5A*), makes a complex with other all known IFT complex B proteins (*Figure 4C*; *Supplementary file 1A*), and undergoes IFT at the same speed as other IFT proteins (*Figure 4A,B*). Nonetheless, IFT56 does not seem to be necessary for assembly or movement of IFT complexes (*Figures 5–7*). Previous studies suggested that *C. elegans* DYF-13 is an IFT regulator that modulates either the activity of the OSM-3-kinesin motor or its association with IFT subcomplex B (*Ou et al., 2005*, *2007*). We speculate that IFT56 is a peripheral component of IFT complex B rather than a core unit of IFT complex B but is not required for activation of kinesin motors at least in *C. reinhardtii*, based on the normal speed of IFT movement in the *dyf13* mutant. Because IFT56 is not required for assembly of IFT complexes, we were able to perform further analyses to clarify the specific function of IFT56, which would have been impossible if complex B failed to form in the *dyf13* mutant cells.

## IFT56 is important to assemble full-length cilia/flagella

We found that IFT56 was important to assemble full-length cilia/flagella. In our study, zebrafish *ttc26* knockdown embryos and *C. reinhardtii* *dyf13* mutant cells could assemble cilia/flagella but they were shorter than control cilia/flagella (*Figures 1 and 2*). These observations coincide with *C. elegans* *dyf-13* mutant, which has abnormal cilia with short length (*Blacque et al., 2005*). Most other IFT proteins are necessary to assemble cilia/flagella in the first place. For example, *C. reinhardtii* mutants of *ift46*, *ift52,* and *ift88* cannot assemble their flagella (*Pazour et al., 2000*; *Brazelton et al., 2001*; *Deane et al., 2001*; *Hou et al., 2007*) because these IFT proteins are necessary to assemble IFT complex B (*Hou et al., 2007*; *Lucker et al., 2010*). The only exception is IFT25 which is not required for assembly of cilia/flagella (*Keady et al., 2012*). Because IFT25 is not conserved in *C. elegans* and *D. melanogaster*,

it might be reasonable that IFT25 is not required for assembly of cilia and IFT complex. In contrast, because IFT56 is conserved in most ciliated organisms (*Figure 1A*) like other IFT proteins, IFT56 should have a conserved role in cilia/flagella or IFT. Further analysis is necessary to clarify the role of IFT56/TTC26/DYF13 in determining the length of cilia and flagella.

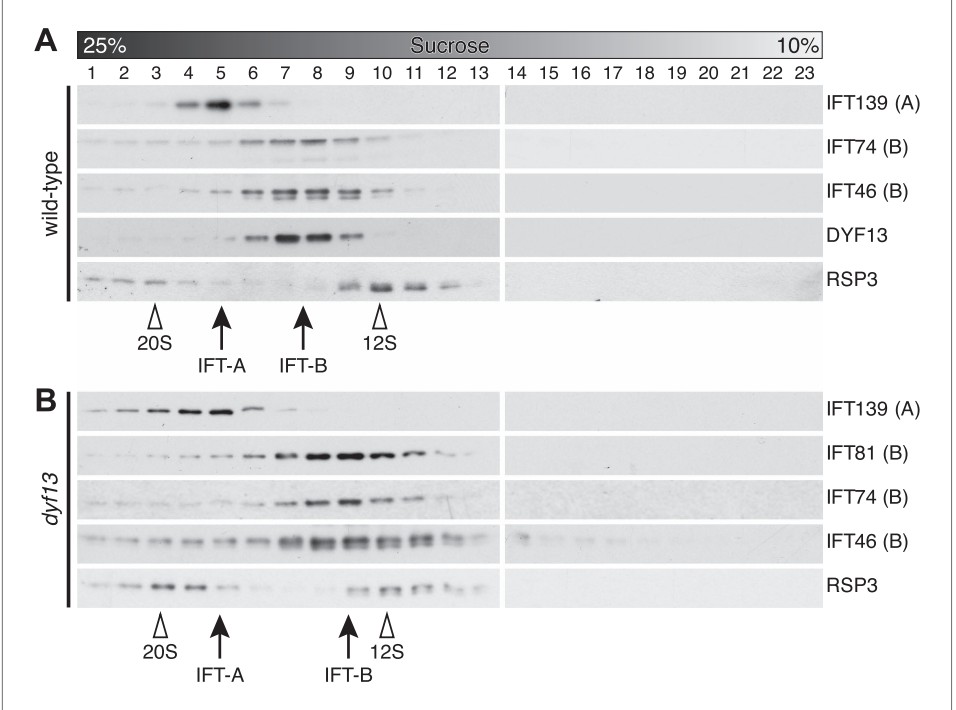

**Figure 5**. DYF13 is a part of IFT complex B. (**A**) The sucrose density gradient fractions of wild-type flagella. Flagellar matrix was fractionated through a 10–25% sucrose density gradient. The sucrose density gradient fractions were separated by 10% SDS-PAGE and analyzed by Western blotting. DYF13 had a peak that coincides with other IFT complex B proteins (IFT74 and IFT46). (**B**) Sucrose density gradient fractions of *dyf13* mutant flagella. The peaks of IFT complex B proteins in *dyf13* mutant flagella shifted to light density in the gradient (B; IFT81, IFT74, and IFT46), but the peak of IFT complex A protein (IFT139) did not shift. RSP3, a component of the radial spoke and not part of the IFT complex, is used as a gradient marker, which sediments at 20S and 12S in the gradient (indicated by white arrowheads). The peaks of IFT complex A and B proteins are indicated by arrows.

## IFT56 is required for transport of a subset of flagella proteins into flagella

Knockdown of IFT56 in zebrafish embryos and *C. reinhardtii dyf13* mutant cells showed motility defects of cilia/flagella. These motility defects are caused by lack of inner dynein arm components in cilia/flagella. IFT56 is not important to assemble IFT complexes but is important to transport inner dynein arms into cilia/flagella. Analysis of a suppressor mutant of *ift46* has revealed that IFT46 is required for transport of outer dynein arms into flagella (*Hou et al., 2007*; *Ahmed et al., 2008*). Some cilia/flagella proteins are known to be pre-assembled before entering the cilia/flagella (*Qin et al., 2004*), including dynein arms, whose pre-assembly requires HSP90-interacting PIH proteins (*Omran et al., 2008*; *Yamamoto et al., 2010*). In *C. reinhardtii*, three PIH proteins, PF13/KTU, MOT48, and TWI1, were identified (*Yamamoto et al., 2010*). PF13/KTU is required for pre-assembly of outer dynein arms (*Omran et al., 2008*) and MOT48 is important to pre-assemble inner-arm dyneins b, c, d, and e (*Yamamoto et al., 2010*). TWI1 might be important to pre-assemble the other dynein arms, inner dynein species a, f, and g. Because *dyf13* mutant cells partially lack inner dynein arms a, f, and g from their flagella, we speculate that IFT56 might play a role in recruiting TWI1 with these inner dynein arms to the IFT complex. Knockdown of *twister*, the zebrafish homologue of TWI1, results in pronephric cysts (*Sun et al., 2004*), similar to the phenotypes of IFT56/TTC26/DYF13 knockdown zebrafish (*Figure 2—figure supplement 1*). We also note the depletion of inner dynein arm components is consistent with the fact that IFT is necessary for incorporation of inner dynein arms but not outer dynein arms in Chlamydomonas (*Piperno and Mead, 1997*). IFT56 is also important for transporting other flagella proteins into flagella (*Figure 8C*; *Supplementary file 1B*) and most of these appear to be related to

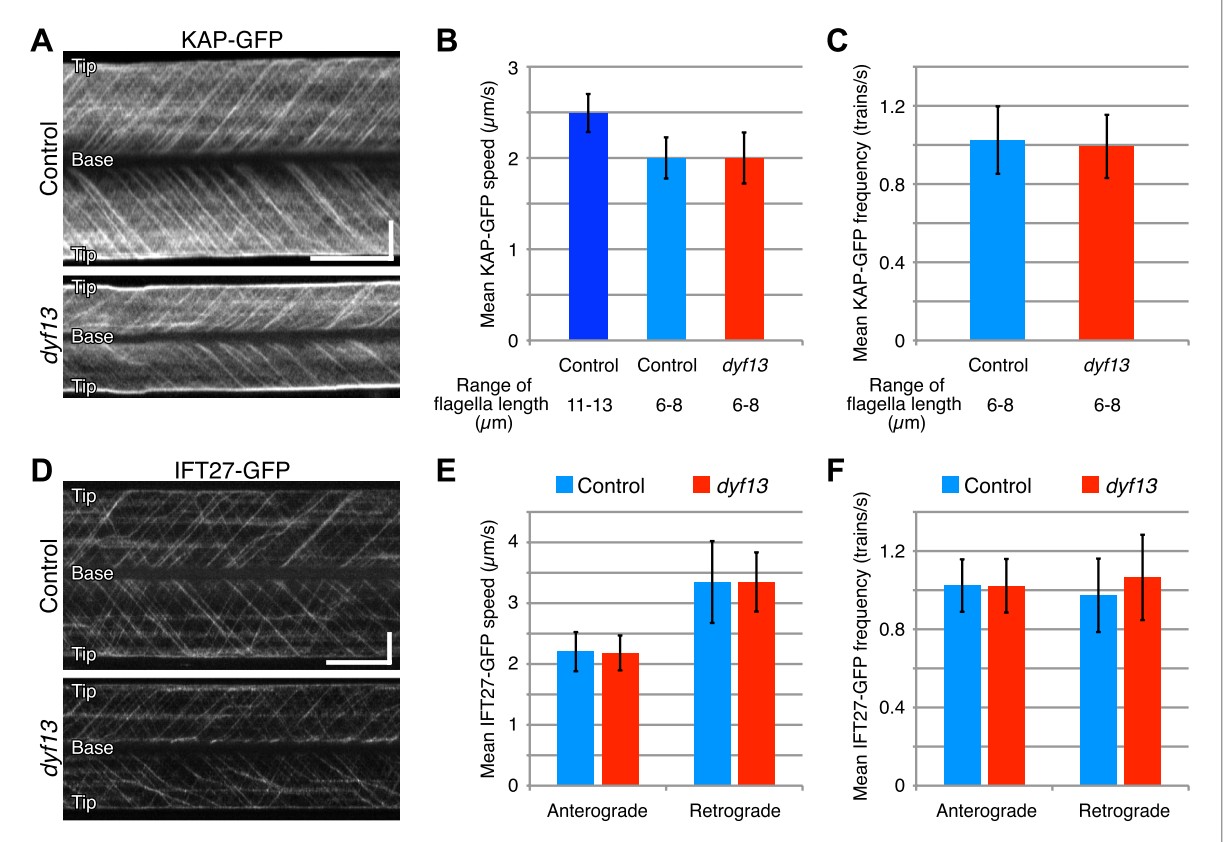

**Figure 6**. DYF13 does not influence IFT particle movement. (**A**) Kymographs of KAP-GFP in control and dyf13 mutant *C. reinhardtii* cells. These kymographs were assembled from **Videos 9 and 10**. Bars: 5 s (horizontal), 5 μm (vertical). (**B**) The mean IFT speeds of KAP-GFP in control and *dyf13* mutant. (**C**) The mean frequencies of KAP-GFP in control and *dyf13* mutant in the same range of flagella length. The mean speeds and frequencies of KAP-GFP in control (light blue) and *dyf13* mutant (red) were almost same in the same range of flagella length (6–8 μm). (**D**) Kymographs of IFT27-GFP in control and *dyf13* mutant *C. reinhardtii* flagella. These kymographs were assembled from **Videos 11 and 12**. Bars: 5 s (horizontal), 5 μm (vertical). (**E**) The mean IFT speeds of IFT27-GFP in control and *dyf13* mutant in the same range of flagella length. (**F**) The mean frequencies of KAP-GFP in control and *dyf13* mutant in the same range of flagella length. The mean speeds and frequencies of IFT27-GFP in control (light blue) and *dyf13* mutant (red) flagella were almost same in the same range of flagella length in both anterograde and retrograde. Error bars represent standard deviations.

flagella motility. FAP59 is a homologue of CCDC39 that is required for assembly of inner dynein and dynein regulatory complexes (*Merveille et al., 2011*). Tektin (*Tanaka et al., 2004*; *Yanagisawa and Kamiya, 2004*) and centrin (*Piperno et al., 1990*, *1992*; *Kagami and Kamiya, 1992*) are both involved in inner dynein arm function and assembly, whereas PF2 (also known as DRC4) is a component of the dynein regulatory complex that regulates inner dynein arm activity (*Rupp and Porter, 2003*). Enolase, a component of the glycolytic pathway, is present in motile flagella as part of the CPC1 central pair complex and a mutant with reduced enolase in flagella causes reduced flagella beat frequency (*Mitchell et al., 2005*). Thus, these proteins depleted from flagella in the *dyf13* mutant are all connected to flagellar motility, rather than flagellar assembly, suggesting a primary role for IFT56/TTC26/DYF13 in carrying motility-related cargoes. Consistent with this picture, IFT56 is not conserved in *Thalassiosira*

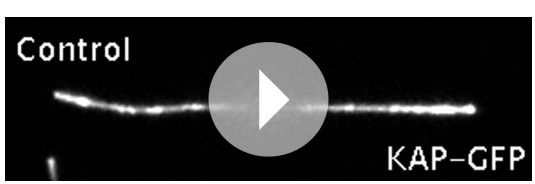

**Video 9**. Fluorescent videomicroscopy of KAP-GFP in *C. reinhardtii* control cell. Images were collected at 20 fps for a total duration of 20 s. Playback is set at 20 fps (real-time speed). KAP-GFP moves toward the tip of flagella in the control *fla3* mutant cell.

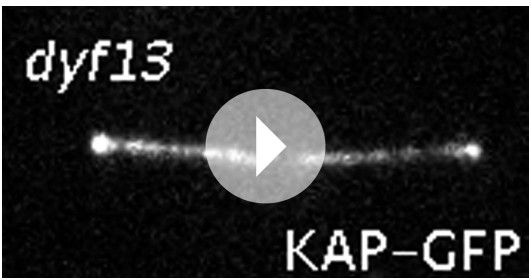

**Video 10**. Fluorescent videomicroscopy of KAP-GFP in *C. reinhardtii dyf13* mutant cell. Images were collected at 20 fps for a total duration of 20 s. Playback is set at 20 fps (real-time speed). KAP-GFP moves toward the tip of flagella in the *dyf13* mutant cell.

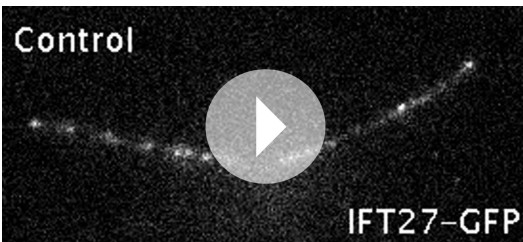

**Video 11**. Fluorescent videomicroscopy of IFT27-GFP in *C. reinhardtii* control cell. Images were collected at 20 fps for a total duration of 25 s. Playback is set at 20 fps (real-time speed). IFT27-GFP moves toward the tip of flagella in the control *pf18* mutant cell.

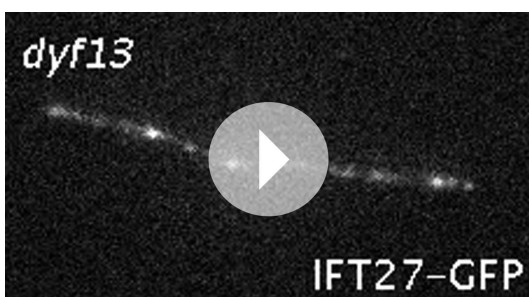

**Video 12**. Fluorescent videomicroscopy of IFT27-GFP in *C. reinhardtii dyf13* mutant cell. Images were collected at 20 fps for a total duration of 25 s. Playback is set at 20 fps (real-time speed). IFT27-GFP moves toward the tip of flagella in the *dyf13* mutant cell.

*pseudonana* or *Physcomitrella patens*, which lack putative inner dynein and outer dynein homologues, respectively (*Wickstead and Gull, 2011*). A subset of proteins was identified as potential cargos of IFT56 in this study. IFT56 might work as an adaptor between IFT main complex and cargos. However, it is still unknown how IFT56 binds with its cargos. Although tubulin association with the IFT complex is mediated directly by tubulin-binding domains on the IFT proteins IFT74 and IFT81 (*Bhogaraju et al., 2013*), we do not at this point know whether IFT56 contains similarly specific binding sites for all of the cargos whose transport it mediates, or whether it might engage additional adaptors as an intermediate stage. Because cargo proteins in some cases form pre-assembled complexes prior to flagellar transport (*Qin et al., 2004*), it may not be necessary for all IFT56 dependent cargos to bind directly to IFT56, because in principle it could be sufficient for a single component of the complex to bind. Also, because IFT56 was found in nonmotile primary and sensory cilia (*Blacque et al., 2005*; *Ishikawa et al., 2012*), which do not have motility complexes, IFT56 likely plays a role in transporting other ciliary components besides motility-related cargoes. For example, since IFT56 is important to assemble full-length cilia in several organisms (*Figure 2D,G and 3F*) (*Blacque et al., 2005*; *Zhang et al., 2012*), it might carry one or more cargoes involved in ciliary length control. Further work will be necessary to determine how IFT56 affects transport in nonmotile cilia.

## Conclusions

In this study, we demonstrated that the mouse cilia proteome candidate TTC26 is an IFT complex B protein with a specific role in transporting a subset of ciliary cargoes related to ciliary motility and for assembling cilia of full length.

## Materials and methods

### Antibodies

Rabbit polyclonal mouse TTC26 and *C. reinhardtii* DYF13 antibodies were raised against a bacterially expressed polypeptide corresponding to residues 312–427 of mouse TTC26 and synthesized peptide of N-terminal 15 amino acids of *C. reinhardtii* DYF13 (MFYSKSRPQHAARTN; Pocono Rabbit Farm and Laboratory, Canadensis, PA). Commercial antibodies, anti-acetylated tubulin (6-11B-1), α-tubulin (DM1A), and actin (AC-15) were purchased from Sigma (St Louis, MO). Antibodies against *C. reinhardtii* proteins IFT22, IFT27, IFT46, IFT70, IFT74/72, IFT81, IFT122, IFT139, IFT140, IFT172, KAP, and D1BLIC were previously reported (*Cole et al., 1998*; *Hou et al., 2004, 2007*; *Qin et al., 2004, 2007*; *Mueller et al., 2005*; *Fan et al., 2010*; *Behal et al., 2012*; *Silva et al., 2012*).

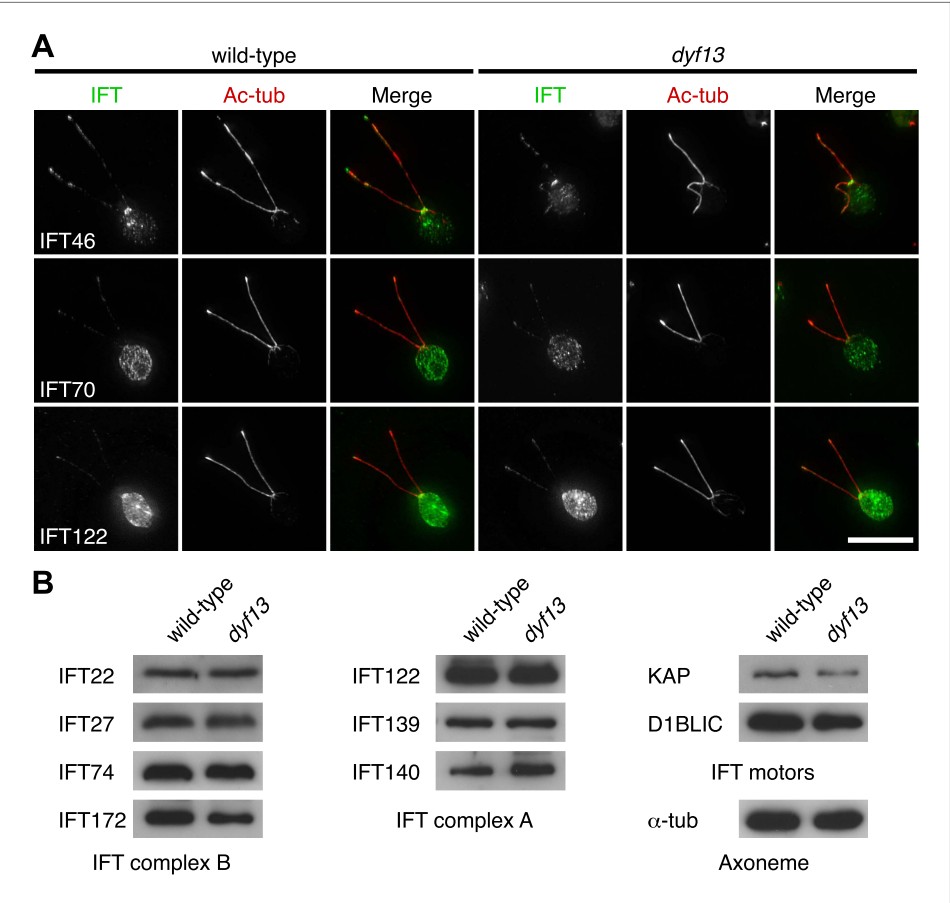

**Figure 7**. DYF13 does not influence localizations and amounts of other IFT proteins. (**A**) Immunofluorescence images of wild-type and *dyf13* mutant cells. The cells were stained with antibodies to acetylated tubulin (red) and IFT46, IFT70, and IFT122 (green). Bar: 10 μm. (**B**) Western blotting images of IFT proteins in wild-type and *dyf13* mutant flagella. Relative amounts of IFT proteins did not change between wild-type and *dyf13* mutant flagella.

## Plasmids

cDNAs of TTC26 and IFT88 were amplified from the first strand cDNA of IMCD3 cells by PCR. TTC26 and IFT88 were subcloned into pCAGGS with C-terminal GFP-tag to generate TTC26-GFP and IFT88-GFP, respectively. TTC26 was also subcloned into pgLAP2 (Addgene, Cambridge MA), which has N-terminal FLAG and S-protein tags, to generate TAP-TTC26.

## Zebrafish morpholino injection

Wild-type TL zebrafish were maintained and raised as described (**Westerfield, 2000**). Embryonic stages were according to somite number, hours post-fertilization (hpf), or days post-fertilization (dpf; **Westerfield, 2000**). Embryos were injected at the one- to two-cell stage with 2–4 ng of morpholinos. The following previously published translational (5'-CTGGCTTCATCCGAGACAAGAGCAT-3') and splice blocking (5'-ATATGTTGGTTCTGATGCACCTGTT-3') morpholinos were injected at the indicated doses (**Zhang et al., 2012**). Negative control morpholino (5'- CCTCTTACCTCAGTTACAATTTATA -3') was also injected for injection control. 1-phenyl-2-thiourea (Sigma) was used to suppress pigmentation when necessary (**Westerfield, 2000**).

## Whole-mount immunofluorescence microscopy

Dechorionated zebrafish embryos at the indicated time points were fixed in 4% paraformaldehyde (PFA; Electron Microscopy Sciences, Hatfield, PA) overnight at 4°C. Fixed samples were washed with PBS, manually deyolked with forceps, and then fixed with Dent's fixative (80% methanol: 20% dimethyl

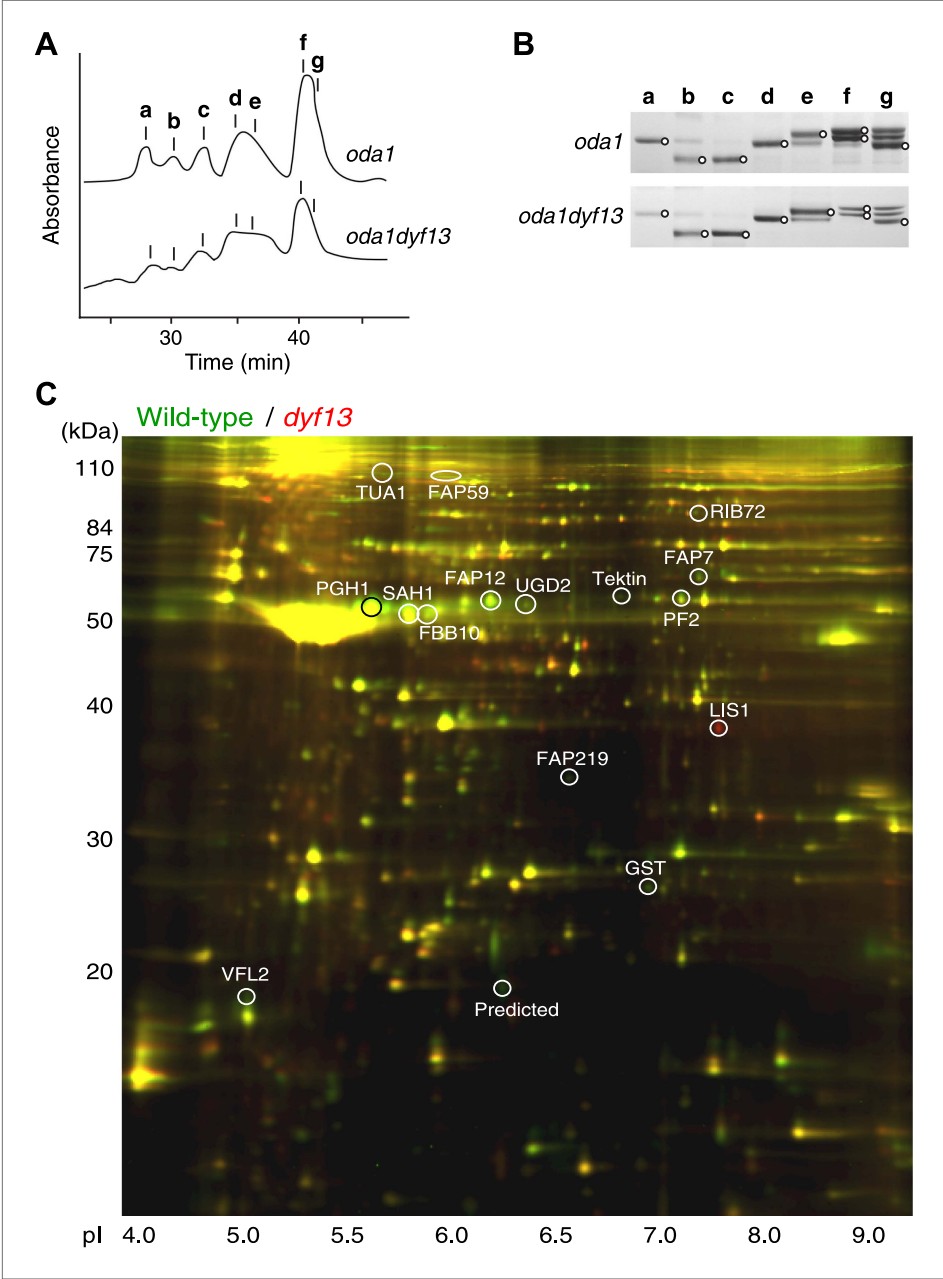

**Figure 8**. *dyf13* mutant reduces the levels of inner dynein arm components in its flagella. (**A**) Elution patterns of inner-arm dyneins of *oda1* single and *oda1dyf13* double mutant *C. reinhardtii* cells. Inner-arm dyneins were extracted with high salt from outer-armless axonemes of *oda1* and *oda1dyf13* cells and fractionated by ion-exchange chromatography on a Mono Q column. The elution positions of each dyneins are indicated by a–g. (**B**) SDS-PAGE of peak fractions showing the dynein heavy chain bands (circles). Peak fractions were subjected to SDS-PAGE with a 3–5% acrylamide gradient and a 3–8 M urea gradient. Dynein a and f were reduced in *oda1dyf13* compared with *oda1*. (**C**) 2D-DIGE proteomic analysis of wild-type and *dyf13* mutant flagella. Fluorescence image of the 2D-DIGE analytical gel, using isoelectric focusing (IEF) in the first dimension and SDS polyacrylamide gel electrophoresis (SDS-PAGE) in the second dimension. Wild-type is shown in green and *dyf13* mutant is shown in red.

The following figure supplements are available for figure 8:

**Figure supplement 1**. *dyf13* mutant does not change the levels of outer dynein arm components in its flagella.

sulfoxide) for 4 hr at room temperature. After fixation, embryos were washed three times with PBS and blocked with 10% goat serum in PBS and 0.1% Tween-20 (PBT) overnight at 4°C. Samples were then incubated with anti-acetylated tubulin antibody (1:1000) in blocking solution overnight at 4°C, washed three times in PBT, and incubated with secondary antibodies (1:200, anti-mouse Alexa 594, Invitrogen, Carlsbad, CA) overnight at 4°C. After extensive PBT washes, embryos were mounted in Vectashield medium (Vector Laboratories, Burlingame, CA). All images and z-stacks were captured using a spectral confocal microscope (Eclipse FN1, Nikon, Tokyo, Japan) with a 60× oil objective.

## KV fluid flow and cilia motility analyses

KV fluid flow and cilia motility analyses were performed as described previously (*Yuan et al., 2012*). Fluorescent red beads (0.5 µm; Invitrogen) were microinjected into the KV and fluorescent imaging was performed on an inverted microscope (Axiovert 200M, Zeiss, Oberkochen, Germany) with a 40× objective and a CCD camera (Axiocam MRm, Zeiss) at five frames per second (fps). Fluorescent bead tracking was performed in ImageJ (NIH, Bethesda, MD). For imaging KV cilia, differential interference contrast (DIC) imaging was performed on an inverted microscope (Axiovert 200M, Zeiss) with a 63 × water objective and a high-speed camera (Phantom Miro, Vision Research, Wayne, NJ) at 1000 fps. Cilia beating dynamics were further analyzed and quantified by generating kymographs of individual motile cilia from a reconstructed time series in ImageJ (NIH).

## Electron microscopy

For transmission electron microscopy of zebrafish retina, 4 dpf zebrafish embryos were doubly fixed with 2% glutaraldehyde in 0.1 M phosphate buffer and 1% osmium tetroxide in 0.1 M phosphate buffer and dehydrated with a graded series of ethanol. Embryos were then embedded in epoxy resin. Ultrathin sections were stained with uranyl acetate and lead citrate and observed with a transmission electron microscope (Hitachi H-7500, Hitachi, Tokyo, Japan) at 80KV with a CCD camera (Advantage 12HR, AMT, Woburn, MA).

## Cell culture and immunofluorescence microscopy

Mouse IMCD3 cells were maintained in a 1:1 mixture of Dulbecco's modified Eagle's medium (DMEM) and Ham's F12 medium with 10% fetal bovine serum at 37°C in 5% $CO_2$. For facilitating ciliogenesis, cells were cultured with serum-free media 1 day before imaging. Antibody staining of IMCD3 cells and imaging were performed as described previously (*Ishikawa et al., 2012*).

## Live cell imaging and image analysis

For IFT imaging in mammalian cells, expression plasmids encoding GFP-tagged TTC26 and IFT88 were transfected into IMCD3 cells using Lipofectamine 2000 (Invitrogen) and then selected with Geneticin (400 µg/ml; Invitrogen) to establish stable clones. The stable clones were selected according to the intensity and localization of GFP-tagged proteins. TTC26- and IFT88-GFP expressing cells were grown under a transwell cup (Corning, Corning NY). The transwell cups were put on a glass bottom dish and cells were imaged on a TIRF microscope (Eclipse Ti, Nikon) with a 100× oil objective. Cells were imaged at 10 fps with adjusted TIRF. IFT imaging of *C. reinhardtii* cells was performed as described previously (*Engel et al., 2009*). Images were collected at 20 fps. Kymographs were generated with ImageJ (NIH).

## Tandem affinity purification analysis

Tandem affinity purification (TAP) analysis was modified from Nachury's method (*Nachury, 2008*). TAP (mock vector as a control) and TAP-tagged TTC26 stably expressing IMCD3 cells were established using the same procedure as was used to generate TTC26-GFP stable cells. TAP-TTC26 expressing cells were suspended with extraction buffer (50 mM HEPES pH 7.5, 150 mM KCl, 1 mM EGTA, 1 mM MgCl2, 10% Glycerol, 1 mM DTT, and 0.1% NP-40) with 10 µg of cytochalasin D (Sigma) and 1 mM PMSF (Sigma) and then incubated on ice for 10 min. After centrifugation at 20000×*g* for 10 min, supernatants were mixed with anti-FLAG M2 affinity gel (Sigma) and incubated 4 hr at 4°C. The affinity gels were washed three times with extraction buffer and S-protein-tagged TTC26 complex was cleaved off FLAG by TEV protease. Eluted S-protein-tagged TTC26 complex was mixed with S-protein agarose (Bethyl laboratories, Montgomery, TX) and incubated overnight at 4°C. TTC26 conjugated S-protein agarose was transferred to a column and TTC26 complex was eluted with 1× SDS sample buffer. The

elution was electrophoresed by SDS-PAGE and the gel was stained with Simply Blue stain (Invitrogen). Bands were cut out and analyzed by mass spectrometry at the Mass Spectrometry and Proteomic Resource Core (Harvard University, Cambridge, MA).

### *C. reinhardtii* strains and culture

The strains used were *C. reinhardtii* wild-type 137c, *oda1* (lacking outer-arm dynein; *Kamiya, 1988*), KAP-GFP *fla3*, and IFT27-GFP *pf18* (*Engel et al., 2009*). The *dyf13* mutant was obtained by UV-mutagenesis and screening for slow-swimming cells. The original mutant having a slow-swimming and short-flagella phenotype was backcrossed with the wild-type three times. In each cross, the same slow-swimming and short-flagella phenotypes segregated 2:2. The resultant progenies of plus and minus mating types were used for experiments. The mutation was identified by positional cloning using amplified fragment length polymorphism (AFLP) analysis of the progenies in 48 tetrads produced by mating with S1-D2 strain (*Kathir et al., 2003*), followed by gene sequencing. In all 48 tetrads the flagellar swimming and length phenotypes were indistinguishable in the mutant products and normal in the other products. Cells were grown in liquid Tris-acetic acid-phosphate (TAP) medium with aeration on a cycle of 12 hr of light and 12 hr of darkness. KAP-GFP *fla3dyf13* double and IFT27-GFP *pf18dyf13* double mutant strains were generated through crosses.

### Flagella motility and swimming speed analyses in *C. reinhardtii*

Flagellar motility analysis was performed as described previously (*Yuan et al., 2012*). Briefly, for imaging *C. reinhardtii* flagella, DIC imaging was taken on an inverted microscope (Axiovert 200M, Zeiss) with a 40× objective and a high-speed camera (Phantom Miro, Vision Research) at 1000 fps. Quantification of flagellar beat frequency and swim speed were performed in ImageJ (NIH).

### Flagella isolation and sucrose density gradient analysis

*C. reinhardtii* flagella isolation and sucrose density gradient analysis were following the same methods as described (*Wang et al., 2009*).

### Dynein composition analysis

Preparation of high salt extract of axonemes and purification of dyneins were carried out according to the method of Kagami and Kamiya (*1992*). Briefly, axonemes of the *oda1* and *oda1dyf13* mutants were suspended in HMDE solution (30 mM HEPES, pH 7.4, 5 mM MgSO$_4$, 1 mM DTT, 1 mM EGTA) containing 0.6 M NaCl and precipitated by centrifugation. The supernatants, referred to as crude dynein extracts, were fractionated into dynein species by high-pressure liquid chromatography on a Mono Q column (Mono Q 5/50 GL, GE Healthcare Bioscience, Tokyo, Japan). The compositions of dynein heavy chains were analyzed by SDS-PAGE with a 4% polyacrylamide and a 6 M urea gel. The gel was stained with silver, and the intensity of dynein bands was analyzed by ImageJ (NIH). We performed this dynein composition analysis twice for *dyf13* vs wild-type and five times for *dyf13oda1* vs *oda1* mutant cells, in all cases obtaining similar results.

### 2D-DIGE analysis

2D-DIGE analysis was performed twice by Applied Biomics (Hayward, CA) as described previously (*Engel et al., 2012*), using wild-type and *dyf13* flagella. Isolation of wild-type and *dyf13* flagella each were done twice, using different cultures grown on separate days, and analyzed entirely independently by DIGE to obtain a measure of variability between samples, as described in the 'Results'. The spots chosen for protein identification by mass spectrometry reported in *Supplementary file 1B* were ones that showed abundance changes of the same sign in both experiments. To allow comparison of ratios between the two experiments, spot volume data for the two fluorescence channels were first normalized by median normalization (*Keeping and Collins, 2011*) using the median values for the first experiment as a reference.

### Acknowledgements

We thank members of the Marshall laboratory for helpful discussions and careful reading of the manuscript and Kurt Thorn for invaluable microscopy resources and assistance. We also thank Takuya Sakaguchi, Shiaulou Yuan, Zhaoxia Sun, Jeremy Reiter, Narendra Pathak, and Iain Drummond for technical advice on zebrafish experiments and analyses and Ryosuke Yamamoto for a suggestion about dyneins. We are grateful to Douglas Cole, Mary Porter, and George Witman for generously sharing antibodies.

# Additional information

### Funding

| Funder | Grant reference number | Author |
|---|---|---|
| National Institutes of Health | GM097017 | Hiroaki Ishikawa, Kimberly A Wemmer, Wallace F Marshall |
| National Science Foundation | MCB-0923835 | Xue Jiang, Hongmin Qin |
| National Institutes of Health | HL54737 | Didier YR Stainier |
| Japan Society for Promotion of Science | 23570189 | Masafumi Hirono, Ritsu Kamiya |
| WM Keck Foundation | | Hiroaki Ishikawa, Wallace F Marshall |
| Packard Foundation | | Didier YR Stainier |
| National Institutes of Health | GM077004 | Wallace F Marshall |
| Japan Society for Promotion of Science | Postdoctoral Fellowship | Hiroaki Ishikawa |
| Herbert W Boyer Postdoctoral Fellowship | | Hiroaki Ishikawa |

The funders had no role in study design, data collection and interpretation, or the decision to submit the work for publication.

### Author contributions

HI, MH, HQ, Conception and design, Acquisition of data, Analysis and interpretation of data, Drafting or revising the article; TI, TY, XJ, HS, HY, KAW, Acquisition of data, Analysis and interpretation of data; DYRS, Conception and design, Drafting or revising the article; RK, WFM, Conception and design, Analysis and interpretation of data, Drafting or revising the article

### Ethics

Animal experimentation: This study was performed in strict accordance with the recommendations in the Guide for the Care and Use of Laboratory Animals of the National Institutes of Health. All of the zebrafish procedures were performed according to approved institutional animal care and use committee (IACUC) protocols of the University of California San Francisco, IACUC protocol approval number AN084150-02.

# Additional files

### Supplementary files

• Supplementary file 1. Results of proteomic analyses. (**A**) List of identified IFT proteins from tandem affinity purification of TTC26. (**B**) List of identified proteins from 2D-DIGE proteomic analysis of wild-type and *dyf13* mutant flagella. Ratios 1 and 2 indicate the fold change in protein abundance quantified in each spot, with negative ratio indicating a reduction in quantity relative to wild-type flagella in each experiment.

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
