## [Decision Letter]

Thank you for sending your work entitled “TTC26/DYF13 is an intraflagellar transport protein required for transport of motility-related proteins into flagella” for consideration at *eLife*. Your article has been favorably evaluated by a Senior editor, a Reviewing editor, and 3 reviewers.

The Reviewing editor and the reviewers discussed their comments before we reached this decision, and the Reviewing editor has assembled the following comments to help you prepare a revised submission.

Three reviewers have seen your paper and all are enthusiastic about your work. Your work provides a nice blend of genetics, imaging and biochemistry, it is of high quality, and interesting. In our discussions, we have come up with two substantive points that we would like you to address before we agree to publication.

1) The proteomic analysis. The quality of the analysis is essential for the conclusions that you draw from the paper. However as far as we can see, you have only done the analysis once. We feel that such a proteomic experiment should be repeated twice, and preferably three times to average the data. We would like to see the analysis repeated, or provided if you have already performed it.

2) We were surprised that the frequency of IFT trains was not reported since a possible reduction in the number of trains could explain the shorter length of the flagellum, at least in *Chlamydomonas*. This is also a feasible experiment given the quality of the kymographs shown at Figure 5. Your group has measured IFT frequency using the same reporter in the Engel JCB2009 paper.

The reviewers were of the opinion that TEM could help interpretation of your data and that if you already have them; this would strengthen the paper. However, we decided that this was not a condition of publication, and leave this up to you.

---

## [Author Response]

*1) The proteomic analysis. The quality of the analysis is essential for the conclusions that you draw from the paper. However as far as we can see, you have only done the analysis once. We feel that such a proteomic experiment should be repeated twice, and preferably three times to average the data. We would like to see the analysis repeated, or provided if you have already performed it*.

This is a very interesting suggestion as we know any proteomic analysis will always have a certain rate of errors, both false positives and false negatives, particularly for proteomic analysis of complex cellular mixtures or organelle preparations. Because of this potential variability, the degree to which one should believe any particular protein identified in the proteomic analysis depends on the reproducibility of the experiment. To address this point, we repeated the entire experiment again, including growing new cultures, isolating flagella, and analyzing them by DIGE. The new results are highly similar to the previous results. In the first analysis, we recognized 2,324 spots in the 2D gel and 64 spots of them were significantly reduced in *dyf13* mutant flagella. The second analysis recognized 3,358 spots and 106 spots of them were reduced. Of the 64 spots reduced in the first analysis, 53 were also reduced in the second analysis, and the ones that were not were all ones that were very close to the cutoff for being classified as having had their abundance changed. More importantly, out of the spots analyzed by mass spectrometry to determine protein identity, all but one (the metabolic enzyme METM) also showed consistent abundance changes in the second analysis, and so our conclusions about the set of proteins depleted in the *dyf13* mutation are supported by this second analysis. We have added all of this information to the Results section, and we provide the ratio data for both experiments in separate columns in Supplementary file 1B. We have noted in the Results section that the abundance changes for these 16 spots are correlated between the two experiments with a correlation coefficient of 0.68, which is statistically significant (p=0.0013). We conclude that while there are differences in the exact number of spots classified as changing in abundance in the two experiments, these differences do not affect the identity of the spots that we actually analyzed. We have also provided more information about the number of experiments done for the dynein arm analysis, which were two experiments for outer dynein arms and five times for inner dynein arms.

*2) We were surprised that the frequency of IFT trains was not reported since a possible reduction in the number of trains could explain the shorter length of the flagellum, at least in* Chlamydomonas*. This is also a feasible experiment given the quality of the kymographs shown at*
Figure 5*. Your group has measured IFT frequency using the same reporter in the Engel JCB2009 paper*.

This is also an extremely good suggestion. Using our previously acquired kymograph data, we measured the frequency of IFT trains in control and *dyf13* mutant cells. The frequency of IFT train did not significantly change in *dyf13* mutant cells. This new data is now presented in Figure 6. We thank the reviewers and editors for insisting on this point, because it supports our conclusion that DYF13 is not necessary for IFT behavior.

*The reviewers were of the opinion that TEM could help interpretation of your data and that if you already have them; this would strengthen the paper. However, we decided that this was not a condition of publication, and leave this up to you*.

We agree that this could be an interesting avenue for future exploration but feel that it is beyond the scope of the present paper.